# Transcriptomic Profiling Reveals an Enhancer RNA Signature for Recurrence Prediction in Colorectal Cancer

**DOI:** 10.3390/genes14010137

**Published:** 2023-01-03

**Authors:** Divya Sahu, Chen-Ching Lin, Ajay Goel

**Affiliations:** 1Department of Molecular Diagnostics and Experimental Therapeutics, Beckman Research Institute of City of Hope Comprehensive Center, Biomedical Research Center, Monrovia, CA 91016, USA; 2Department of Genetics, University of Alabama at Birmingham, Birmingham, AL 35294, USA; 3Institute of Biomedical Informatics, National Yang Ming Chiao Tung University, Taipei 11221, Taiwan

**Keywords:** colorectal cancer, enhancer RNA, recurrence prediction biomarker, transcriptome expression profiling, LASSO Cox regression

## Abstract

Background: Colorectal cancer (CRC) is one of the most fatal malignancies worldwide, and this is in part due to high rates of tumor recurrence in these patients. Currently, TNM staging remains the gold standard for predicting prognosis and recurrence in CRC patients; however, this approach is inadequate for identifying high-risk patients with the highest likelihood of disease recurrence. Recent evidence has revealed that enhancer RNAs (eRNAs) represent a higher level of cellular regulation, and their expression is frequently dysregulated in several cancers, including CRC. However, the clinical significance of eRNAs as recurrence predictor biomarkers in CRC remains unexplored, which is the primary aim of this study. Results: We performed a systematic analysis of eRNA expression profiles in colon cancer (CC) and rectal cancer (RC) patients from the TCGA dataset. By using rigorous biomarker discovery approaches by splitting the entire dataset into a training and testing cohort, we identified a 22-eRNA panel in CC and a 19-eRNA panel in RC for predicting tumor recurrence. The Kaplan–Meier analysis showed that biomarker panels robustly stratified low and high-risk CC (*p* = 7.29 × 10^−5^) and RC (*p* = 6.81 × 10^−3^) patients with recurrence. Multivariate and LASSO Cox regression models indicated that both biomarker panels were independent predictors of recurrence and significantly superior to TNM staging in CC (HR = 11.89, *p* = 9.54 × 10^−4^) and RC (HR = 3.91, *p* = 3.52 × 10^−2^). Notably, the ROC curves demonstrated that both panels exhibited excellent recurrence prediction accuracy in CC (AUC = 0.833; 95% CI: 0.74–0.93) and RC (AUC = 0.834; 95% CI: 0.72–0.92) patients. Subsequently, a combination signature that included the eRNA panels and TNM staging achieved an even greater predictive accuracy in patients with CC (AUC = 0.85). Conclusions: Herein, we report a novel eRNA signature for predicting recurrence in patients with CRC. Further experimental validation in independent clinical cohorts, these biomarkers can potentially improve current risk stratification approaches for guiding precision oncology treatments in patients suffering from this lethal malignancy.

## 1. Introduction

Despite advances in the diagnosis and treatment of colorectal cancer (CRC), the disease still accounts for approximately 10% of global cancer cases [1]. According to the American Cancer Society, an estimated 52,580 deaths are expected from CRC in the United States in 2022 [2]. CRC is a highly heterogeneous disease [3]. Depending on the anatomical location where the disease initiates, it can be classified into colon cancer (CC) or rectal cancer (RC); however, both disease types are often grouped because they share similar molecular characteristics [4]. The five-year survival rate for CRC patients is approximately 65% for all stages [2]. However, only an estimated 38% of patients are diagnosed with localized disease, for which the five-year survival rate can be more than 90% [2]. However, anatomically, the five-year survival rate for all stages in CC is 63% and for RC is 67% [2]. In the current clinical practice, clinicopathological features such as patient age, gender, primary site location, tumor grade, and tumor-node-metastasis (TNM) staging are used for predicting prognosis and recurrence in CRC patients [5]. However, these conventional clinical practices are not precise, resulting in inaccurate patient stratification [6,7]. Several protein-coding genes have been developed as molecular markers for CRC prognosis [8,9,10,11], yet the survival rate is unsatisfactory [12]. Therefore, there is imperative to discover novel recurrence-associated biomarkers that may contribute to improved risk stratification and identify high-risk CRC patients for appropriate clinical decision-making and treatments.

With the completion of the Human Genome Project in 2003, it is evident that protein-coding genes account for merely 1–2% of the human genome [13], while an astounding proportion of the genome serves as a noncoding component, which was previously characterized as a dark matter of the genome. Noncoding RNAs are a rich assortment of RNA molecules spanning from short RNAs (including miRNAs, piRNAs, and snoRNAs) to long RNAs (including lncRNAs and circular RNAs) [14]. Evidence suggests that these noncoding RNAs play a regulatory role in the biological processes and are involved in cancer development and progression [14,15]. Recently, enhancer RNAs (eRNAs) that are between 50–2000 nucleotide long have been discovered as one class of noncoding RNAs that are bidirectionally transcribed from the enhancer regions [16]. Several studies have reported dysregulation of eRNAs in cancers [17], which leads to the altered expression of oncogenes [18,19] and tumor suppressor genes [20]. Gu et al. [21] identified AP001056.1 enhancer RNA high expression was associated with good outcomes in patients with squamous cell carcinoma of the head and neck carcinoma. Likewise, eRNAs regulate active enhancers, promote gene expression and affect cell-specific transcriptional regulation [16,22]. In colon cancer, CRISPR loss-of-function screens analysis identified bromodomain and extra terminal (BET) member protein BRD4. Inhibition of BRD4 decreased tumor growth and increased differentiation of CIMP+ colon cancer tumors. JQ1, a BET inhibitor, reduced colon cancer proliferation. Further examining the RNA-Seq and ChIP-Seq data after JQ1 treatment led to the identification of colon cancer-associated transcript 1 (CCAT1) eRNA, which was significantly downregulated. CCAT1 enhancer regulates cMYC expression and was associated with poor outcomes in colon cancer patients [19]. The expression level of eRNAs within the enhancer regions positively correlates with the expression level of nearby protein-coding genes [16]. Sweeney et al. [23] demonstrated that Kallikrein-related peptidase 3 enhancer RNA (KLK3e) mediates KLK3 enhancer and KLK2 promoter interaction and promotes KLK2 transcription. In addition, KLK3e showed a positive correlation with KLK2 gene expression (neighbor target gene) in both primary and metastatic prostate tumors. This evidence suggests eRNAs can be considered as genome wide markers for active transcriptional regulation and prognostic markers predictive of patient cancer outcome.

In view of the emerging interest for eRNAs as a promising category of biomarkers, herein we developed an unbiased, systematic bioinformatics approach to identify a novel and robust eRNA panel that can predict tumor recurrence in patients with stage II and III colorectal cancer.

## 2. Results

### 2.1. Building Survival Model to Identify Clinically Relevant eRNAs

Through the following steps, we developed an eRNA-based survival model to identify clinically relevant eRNA for CC and RC (Figure 1).

We randomly selected 70% of the entire cohort from The Cancer Genome Atlas (TCGA) as the training set and reserved the remaining 30% of the cohort as the testing set. To avoid deviation affecting the stability of the model, we maintained the distribution of disease-free and relapsed patients from the entire cohort in both training and testing sets.Multivariate Cox regression analysis was carried out training on 477 eRNAs for CC and 460 eRNAs for RC, along with controlling the effects from other clinical risk factors, including patient age, gender, and TNM stage.eRNAs significantly associated (*p* < 0.05 & z-score > 1.96) for predicting patient disease-free survival (DFS) were retained and termed as prognostic eRNAs and were further selected by LASSO Cox regularization with 10-fold cross-validation.Following LASSO Cox regularization and eRNA selection, a risk score formula was established. The risk score for each patient was calculated by a linear combination of expression and multivariate Cox coefficient of eRNAs.Patients were classified into low-risk or high-risk groups using the median risk score as the cut-off threshold from the training set. The coefficient for each eRNA and the cut-off value of the risk score from the training set was used to calculate the risk score and to stratify patients into individual risk groups in the testing set.Survival differences between the two groups were estimated using Kaplan–Meier curve and compared using the log-rank test.

The above-mentioned steps 1–6 were repeated 100 times to obtain 100 different eRNA sets. Only those sets that significantly stratified patients into two groups (high-risk groups exhibited poorer survival than low-risk groups) in the testing were termed successful survival models or failed models. The results obtained from one such trial are shown in Appendix A. For CC, we applied the model to the entire cohort (n = 379), however, the model was generalized, and we found very few prognostic eRNAs signatures, which successfully stratified patients into two different risk groups within the testing cohort. Due to CC heterogeneity, we identified prognostic eRNAs signatures within CC subtypes based on Consensus Molecular Subtypes (CMS) classification. The entire CC cohort (n = 379) was classified into 5 subtypes, with CMS1 comprising 13%, CMS2 23%, CMS3 14%, CMS4 29%, and mixed subtype 21% (Figure 2A). The proportion of patients classified into CMS subtypes was approximately similar to those described in Guinney et al. [24] study. We further checked the five-year DFS probability of each CMS subtype with CMS1: 89%; CMS2: 64%; CMS3: 81%; CMS4: 40%; and mixed subtype: 60%, as shown in Figure 2B. As CMS4 displayed the worst survival compared to other CMS subtypes, we focused on identifying a clinically relevant eRNA signature for this subtype. We found 22 prognostic eRNAs for the CC-CMS4 subtype and 19 prognostic eRNAs for RC that significantly distinguished patients into two risk groups in the testing set (Figure 2C,D).

### 2.2. Prognostic Association of eRNA Signature Risk Score with Disease-Free Survival in Patients

A risk score was constructed using the median Cox-coefficients from the successful survival models. For the CC-CMS4 subtype, compared with patients in the low-risk score group, those with high-risk scores exhibited poor DFS (Figure 3A). Only 17% of CC patients in the high-risk group were disease-free at 5 years, compared with 93% of the patients in the low-risk score group. We further randomly divided the entire cohort into 50% and used the same risk-scoring formula and cut-off thresholds of the whole cohort, wherein the 22-eRNA signature still significantly stratified patients into two risk groups for DFS (Figure 3B). For RC, 54% of the patients in the high-risk score group were disease-free at 5 years, compared with 84% of the low-risk groups (Figure 3C). Similarly, upon randomly splitting the RC cohort into 50% and using the same cut-off threshold, the 19-eRNA signature significantly stratified the patients into two risk groups (Figure 3D).

### 2.3. Recurrence Prediction by eRNA Signature Is Independent of Clinical Risk Factors

To evaluate the prognostic independence of eRNA signature against the known clinical risk factors, multivariate Cox regression analysis revealed that the 22-eRNA signature was an independent predictor of DFS in CC patients (Hazard Ratio [HR] = 11.89, *p* = 9.54 × 10^−4^ Figure 4A). We also found that patients with relapsed tumors manifested significantly higher risk scores than disease-free patients (*p* = 5.72 × 10^−7^; Figure 4B). Similarly, the 19-eRNA signature was independently associated with DFS in RC (HR = 3.91, *p* = 3.52 × 10^−2^; Figure 4C), and their expression was significantly higher in the relapsed patient (*p* = 2.61 × 10^−6^; Figure 4D).

### 2.4. The eRNA Signature Is a Better Predictor of Recurrence with High Sensitivity and Specificity

To confirm the prediction accuracy for DFS, we performed the Receiver Operating Characteristic Curve analysis (ROC) curve analysis for the eRNA signature. In both CC and RC, we observed that the eRNA signature exhibited excellent prediction power for relapse. For CC, the area under the curve (AUC) and 95% confidence interval (CI) for the 22-eRNA signature was 0.83 and 0.739–0.928 (Figure 5A), and for RC, the 19-eRNA signature achieved a remarkable prediction accuracy (AUC = 0.834, 95% CI = 0.737–0.931; Figure 5B). Subsequently, when the eRNA signature was combined with the TNM stage, it achieved an even higher prediction accuracy in CC (AUC = 0.847), suggesting its potential clinical utility in this malignancy. However, no increment in prediction accuracy was observed in RC when combined with the TNM stage.

### 2.5. Higher Expression of eRNAs Associated with Tumor Recurrence Compared to Its Target Genes

We performed the annotation of each prognostic eRNA (as suggested by Zhang et al. [25]) and examined the expression profiles of eRNAs and their neighboring target genes, which were highly positively correlated. For example, *AADAC*-associated eRNA (*AADACe*, ENSR00000160285) was highly associated with the *AADAC* gene in CC (SCC = 0.43, *p* = 1.03 × 10^−5^, as shown in Figure 6A. Similarly, *TRMT6*-associated eRNA (*TRMT6e*, ENSR00000106418) was highly correlated with the expression of the *TRMT6* gene in RC (SCC = 0.38, *p* = 1.53 × 10^−5^) as shown in Figure 6B. Several studies have reported that eRNA expression has a better survival prediction potential than target genes. In support of the previous studies, we also observed that a high level of *AADACe* was associated with worse survival (log-rank test *p* = 0.02058, Figure 6C). Interestingly, the *AADAC* gene was not associated with CC patient survival, suggesting *AADACe* clinical utility regardless of *AADAC* expression in CC patients. Similar results were observed for *TRMT6*-associated eRNA (*TRMT6e*) in RC, as shown in Figure 6D. Further, we compared the prediction accuracy of the eRNA signature with other published CRC recurrence prediction gene signatures [26,27]. We extracted their gene list, performed multivariate Cox regression analysis, built risk scores from the CC cohort, and calculated the AUC of the ROC curve. As expected, the eRNA signature showed higher performance for DFS than other prognostic gene signatures (Appendix A).

### 2.6. Putative Biological Functions of eRNA Signature in Colorectal Cancer

The eRNAs have no protein-coding capacity; thus, we applied a guilt-by-association strategy to explore the putative potential biological functions of the eRNA signature in CRC, as shown in Figure 7A. We found biological functions related to metabolic process, DNA methylation, transcription initiation, cell cycle, and myeloid leukocyte mediated immunity for CC eRNA signature. Similarly, biological functions associated with the regulation of DNA replication, ncRNA metabolic process, mRNA processing, and oxidative phosphorylation were enriched for RC eRNA signature. The highest enriched biological functions for *TATDN2*-associated eRNA (*TATDN2e*, ENSR00000148476) in CC was histone H3K4 methylation, and *CDH17*-associated eRNA (*CDH17e*, ENSR00000227152) in RC was regulation of mRNA splicing by spliceosomes (Figure 7B,C).

## 3. Discussion

The human eRNAs are a specific class of noncoding regulatory RNAs transcribed from the genomic enhancer regions and act in cis-direction to influence the transcription of the neighboring protein-coding genes. To the best of our knowledge, our study is the first to report the analysis and discovery of an RNA-seq-based prognostic eRNA signature in CRC. Cox proportional hazard regression is widely used to decipher the association of molecular marker expression profiles with disease outcomes. Using the eRNA expression profile for the CC and RC cohort from the TCGA database, we identified a 22- and 19-eRNA panel for predicting prognosis in CC and RC, respectively. Using a machine learning approach for the random selection of eRNAs from the training set and further validation of the survival model in the testing set with 100 iterations increased the robustness of the identified prognostic eRNAs. A risk-scoring formula was applied to integrate predictive eRNA expression into a signature. Patients were stratified into a low-risk score group and a high-risk score group based on the median risk score of the entire cohort. Kaplan–Meier survival analysis with Mantel log-rank test evaluated the association of eRNA signature with patient DFS. We validated the prognostic association of the signature by randomly selecting 50% of the cohort and using the median risk score of the entire cohort to stratify the patient into two risk score groups. Multivariable Cox analysis was used to estimate the prognostic independence of the eRNA signature against the established clinical risk factors in CRC. ROC curve analysis demonstrated that the CC recurrence prediction accuracy for the eRNA signature was higher than the protein-coding genes signature.

The human CC is a highly heterogeneous disease; therefore, we identified consensus molecular subtypes using gene sets for consensus molecular subtype classification. As the CMS4 subtype was associated with the worst prognosis, we developed a prognostic signature for this subtype. In this study, we also incorporated stage 1 patients, as we hypothesized that these patients in CMS4 subtypes might have a higher risk of relapse than in other subtypes. We found that among 477 eRNAs from CC and 460 eRNAs from RC, only 412 eRNAs (78%) were in common, suggesting similarity in the eRNA landscape in the two-cancer type, however, we found only 1 prognostic eRNA *ENSR00000021608* similar between the two cancer types (Appendix A) suggesting prognostic specificity of eRNA in their respective cohort.

We would like to acknowledge the potential limitations of our study. First, our study’s CC and RC cohorts were only from the TCGA. Therefore, this study used a random selection of prognostic eRNA signatures from the training set and further validated the successful stratification of the cohort into two risk score groups in the testing set. Second, the mutational status of KRAS and BRAF genes and CIMP and MSI status were not included as a covariate in the multivariate Cox analysis as information for these clinical risk factors were unavailable for these patient cohorts. Third, our study could not significantly stratify the patients into two risk score groups for overall survival because only a few deaths were observed (death vs. censored in CC; 14/82 and in RC; 5/116). Despite these drawbacks, significant stratification of patients into two risk score groups in the testing sets in various iterations and further validation in the entire cohort with sophisticated statistical analysis offers a high confidence level in the overall analyses.

## 4. Conclusions

To conclude, we developed a signature consisting of 22 eRNAs in CC and 19 eRNAs in RC that can predict tumor relapse in these two malignancies. In addition, the expression level of the eRNA signature is sensitive and profiled using the RNA-seq platform. The expression of eRNA signature possesses a superior prediction power for recurrence compared to other clinical risk factors. The eRNA signature was independent in predicting patient tumor relapse. Thus, our study provides novel evidence for recurrence-associated eRNAs for experimental validation in clinical cohorts and deeper investigations to better understand the CRC etiology and progression while providing a clinical application for precise risk stratification to identify those groups of patients at higher risk for CRC recurrence.

## 5. Methods

### 5.1. CRC Patients Datasets

The expression profiles of eRNAs for CC and RC were downloaded from the eRNA data portal in the cancer (eRic) database [25]. Here, RNA-seq expression profiles from the TCGA database were mapped to eRNA regions, followed by calculating their expression levels as reads per million (RPM) for each patient. For CC and RC, we retrieved 477 and 460 eRNA transcripts. The datasets were log2 (RPM + 1) normalized. The clinical information for CRC patients was downloaded from the cBioPortal database [28,29]. Only CRC patients with TNM stages 1, 2, and 3 were included. This study included 96 CC patients with CMS4 subtype and 121 RC patients for which both RNA-seq and survival information were available. The demographics of CC and RC patients are shown in Appendix A.

### 5.2. Consensus Molecular Subtypes (CMS) Classification

The CMS classification for each patient with CC was performed using gene sets with Synapse ID syn4983432 (5973 genes available) downloaded from the Sage-Bionetworks (https://sagebionetworks.org/research-projects/colorectal-cancer-subtyping-consortium-crcsc/ (accessed on 12 July 2020). The R package *CMS classifier* was used for CMS subtype calling using a random forest classifier function with a default posterior probability of 0.5 [24]. Based on the similarity of gene expression profiles in this method, all CC patients were classified into four subtypes CMS1—CMS4. All CC patients with mixed features or intra-tumoral heterogeneity were classified as mixed subtypes.

### 5.3. Identification of Prognostic eRNAs Associated with Tumor Relapse

We randomized 70% of the entire cohort as a training set and the rest 30% as a testing set. The distribution of disease-free and relapse-free patient populations in training and testing sets was comparable. Since clinical information, including patient age, gender, and tumor stage, can affect patient survival, we accordingly examined the prognostic potential of each eRNA using multivariate Cox regression analysis in the training set. We used zscore > 1.96 (which corresponds to *p* < 0.05) as a cut-off threshold to select prognostic eRNAs associated with disease-free survival (DFS). We used the LASSO Cox regression with 10-fold cross-validation to identify a subset of prognostic eRNAs and establish an eRNA signature. The eRNA signature’s prognostic potential was evaluated in the testing set. We repeated the above split cohort approach into a training set, and a testing set 100 times, which generated 100 different sets of eRNA signatures.

### 5.4. Risk Score Calculation

During each iteration, the risk score for each patient was calculated using a linear combination of expression and coefficients from multivariate Cox regression analysis as follows:$$\mathrm{Risk}\;\mathrm{score}=\sum_{j=1}^{n}e_{j}\times exp_{ij},$$
where *e_j_* is the multivariate coefficient of eRNA *j*, *exp_ij_* is the expression value of eRNA *j* in patient *i*, and *n* is the number of prognostic eRNAs (selected after lasso Cox analysis). To calculate the risk score of the prognostic eRNAs chosen from all the successful survival models (which significantly stratify the patients into two groups in the testing set), we used median Cox coefficients obtained from lasso Cox analysis.

### 5.5. Statistical Analysis

Kaplan–Meier survival analysis (low-risk score groups vs. high-risk score groups) with Mantel log-rank test was performed to assess differences between the two survival probabilities. Multivariable Cox proportional hazard regression was performed to determine the prognostic independence of risk scores and clinical risk factors. ROC curve and AUC analyses were performed to evaluate the sensitivity and specificity of eRNA risk scores for predicting DFS. Survival analysis was performed using the *survival* R package [30]. LASSO Cox proportional hazard regression analysis was performed using the *glmnet* R package [31]. ROC curve and AUC analysis were performed using the *pROC* R package [32], and plots were generated using the *ggplot2* R package [33].

### 5.6. Gene Set Enrichment Analysis

Spearman correlation coefficient (SCC) was calculated between prognostic eRNAs and whole genome protein-coding genes (PCGs) in CC and RC patients, respectively. To assess the putative biological functions of each eRNA, gene set enrichment analysis (GSEA) was performed using MSigDB (C5.bp.v7.1.symbols.gmt gene set collection; 7530 gene set available), ranked list of eRNAs correlated genes and their SCC, maximum gene set size of 5000, minimum gene set size of 15, weighted enrichment statistics, and 1000 permutations. GSEA analysis was performed using GSEA pre-ranked tool [34]. Over-represented gene sets (FDR *q*-value 0.05 and overlap coefficient of 0.5) were filtered and visualized using an Enrichment map-a Cytoscape plug-in [35].

## Figures and Tables

**Figure 1 genes-14-00137-f001:**
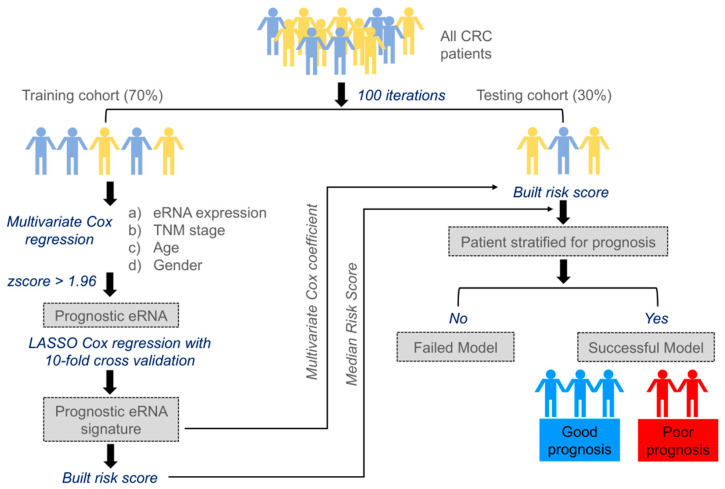
Workflow showing steps involved in the identification of prognostic enhancer RNA signature.

**Figure 2 genes-14-00137-f002:**
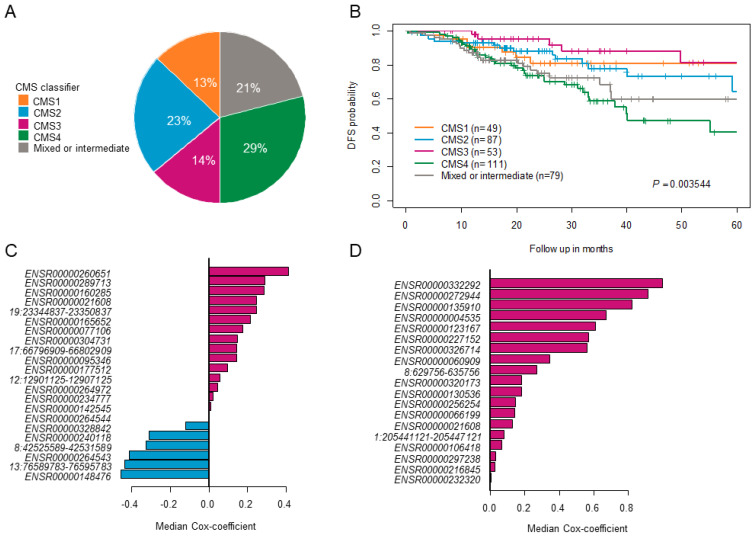
Identification of clinically relevant enhancer RNAs. (**A**) Pie-chart shows the percentage of colon cancer patients classified into various consensus molecular subtypes (CMS). (**B**) Kaplan–Meier plot shows survival estimates of disease-free survival in colon cancer patients stratified based on the CMS classification. (**C**,**D**) The bar graph shows prognostic eRNAs from the successful trials that significantly separated samples into two risk groups in the testing set and are ordered by their median cox-coefficient for disease-free survival. Positive scores are associated with shorter survival, and negative scores are associated with longer survival—22 prognostic eRNAs in colon cancer CMS4 subtype and 19 prognostic eRNAs in rectal cancer. ERNAs = enhancer RNAs.

**Figure 3 genes-14-00137-f003:**
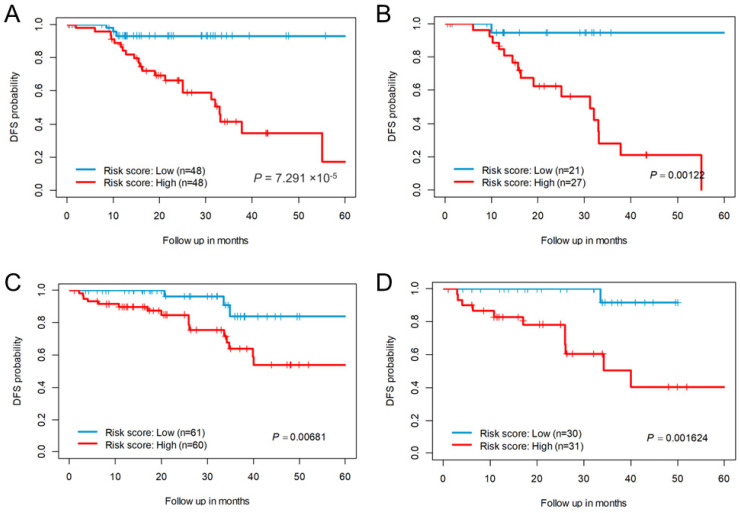
Survival estimates of enhancer RNA signature for disease-free survival probability. (**A**) Kaplan–Meier plots show low- and high-risk score groups based on the median risk score of the colon cancer CM4 subtype cohort (**B**) Survival estimates of randomly selected 50% of the colon cancer CMS4 subtype cohort. Kaplan–Meier plots show low- and high-risk score groups based on the median risk score threshold value from the entire colon cancer CMS4 cohort. (**C**) Kaplan–Meier plots show low- and high-risk score groups based on the median risk score of the rectal cancer cohort (**D**) Survival estimates of randomly selected 50% of the rectal cancer cohort. Kaplan–Meier plots show low- and high-risk score groups based on the median risk score threshold value from the entire rectal cancer cohort. The *p* values were obtained using a Mantel log-rank test (two-sided). DFS = Disease-free survival.

**Figure 4 genes-14-00137-f004:**
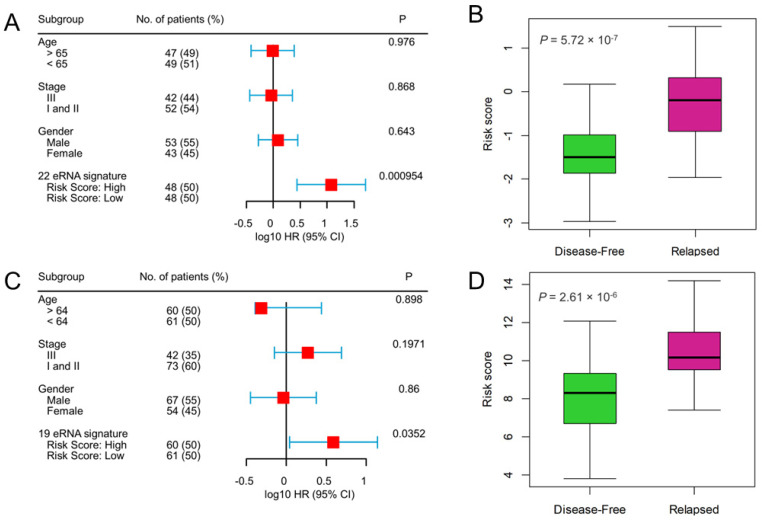
Multivariate Cox-regression analysis shows survival prediction by enhancer RNA signature is independent of clinical risk factors, and enhancer RNA signature risk score is up-regulated in the relapsed patient. (**A**,**B**) colon cancer CMS4 subtype, (**C**,**D**) rectal cancer. A Forest plot of the eRNA signature shows that patients in high-risk score groups had poor outcomes and an independent predictor of disease-free survival after adjusting for the clinical risk factors. The hazard ratio and the confidence interval for colon and rectal cancer are represented on a log10 scale. All statistical tests were two-sided. CI = confidence interval, HR = hazard ratio, eRNA = enhancer RNA.

**Figure 5 genes-14-00137-f005:**
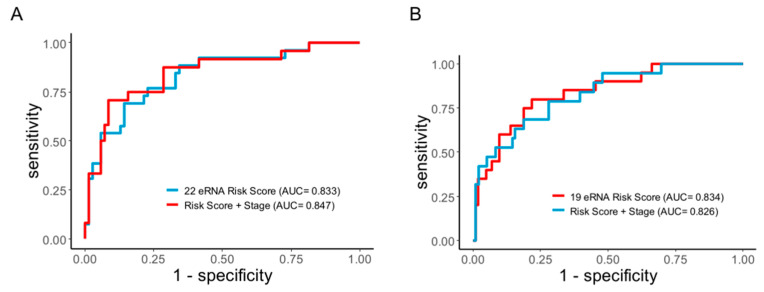
Receiver operating characteristics (ROC) curve analyses of disease-free survival prediction by the enhancer RNA signature. ROC curve shows eRNA higher sensitivity and specificity for predicting disease-free survival prediction when combined with TNM staging (**A**) colon cancer CMS4 subtype (**B**) rectal cancer. eRNA = enhancer RNA; ROC = receiver operating characteristic.

**Figure 6 genes-14-00137-f006:**
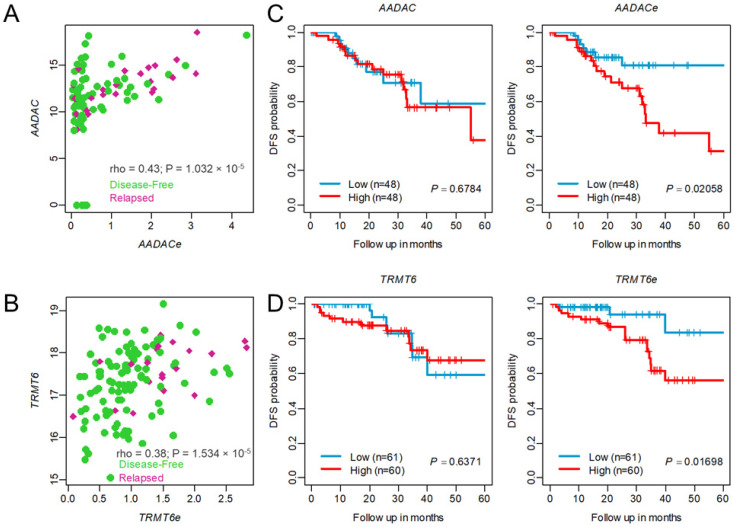
Association between enhancer RNA and their neighbor protein-coding target genes. Spearman correlation coefficient (SCC) analysis shows a positive correlation between enhancer RNA and neighbor protein-coding genes from the (**A**) colon cancer CMS4 subtype cohort and (**B**) rectal cancer cohort. Dots in green represent disease-free patients, and red represents patients with relapsed tumors. (**C**,**D**) Kaplan–Meier plots show enhancer RNA higher expression is associated with tumor relapsed than its neighbor target genes in the colon cancer CMS4 subtype cohort and rectal cancer cohort. rho = spearman correlation coefficient.

**Figure 7 genes-14-00137-f007:**
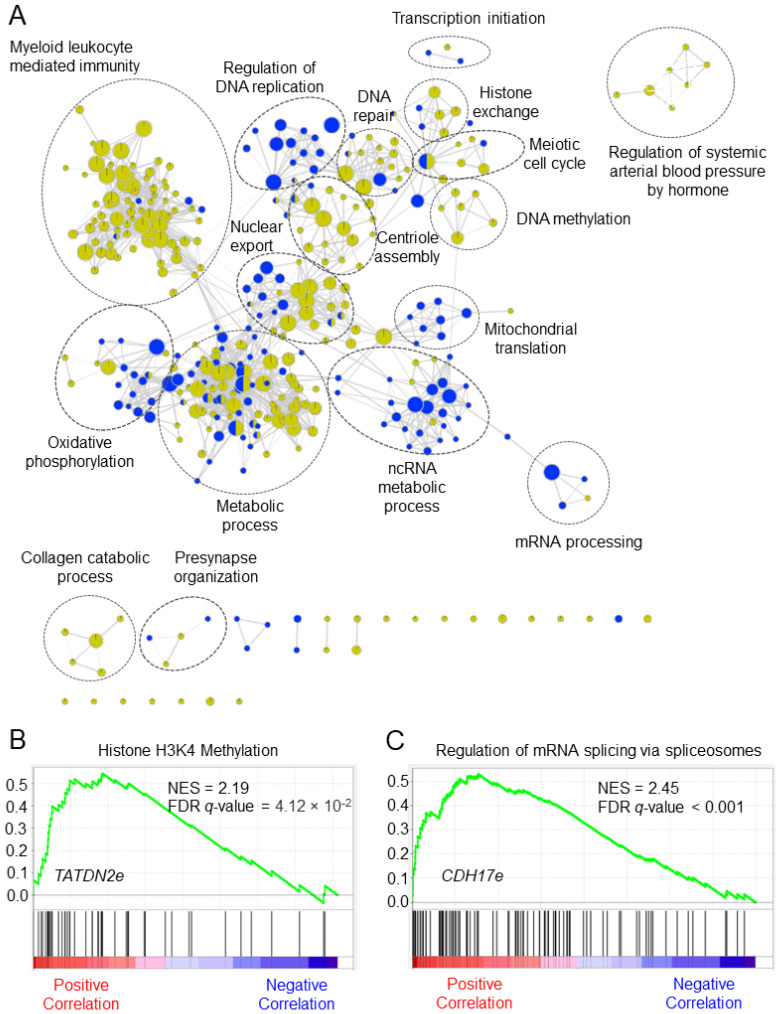
Putative biological functions of the enhancer RNA signature in colorectal cancer. (**A**) The network shows overrepresented gene sets for the signature. Green nodes represent gene sets enriched in colon cancer, and blue nodes represent gene sets enriched in rectal cancer. Node size is proportional to the normalized enrichment score. Genes sets with similar biological functions tend to form clusters, these were manually identified, and gene ontology terms with the highest NES were labeled. (**B**,**C**) The enrichment plot shows the highest enriched function for *TATDN2*-associated eRNA (*TATDN2e*, ENSR00000148476) in colon cancer and *CDH17*-associated eRNA (*CDH17e*, ENSR00000227152). FDR = false discovery rate, NES = normalized enrichment score.

## Data Availability

The eRNA expression dataset used in this study can be found in the eric database https://hanlab.uth.edu/eRic/ (accessed on 15 May 2020). The clinical information for CC and RC can be downloaded from the cBioPortal database https://www.cbioportal.org (accessed on 6 May 2020).

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
