# Peer review of "Transcriptomic Profiling Reveals an Enhancer RNA Signature for Recurrence Prediction in Colorectal Cancer"

_genes, 2023, doi:10.3390/genes14010137_

Round 1
Reviewer 1 Report
- Can you show expression correlation between the master transcription factors in colorectal cancer and the eRNA signature you identified?
- Did you explore the possibility of elucidating eRNA- gene regulatory network in colorectal cancer from High-throughput chromosome conformation capture (Hi-C) data which is available?
- It would be interesting to examine drug-eRNAs interactions from the available databases.
- Include sufficient points in the introduction and discussion suggesting why eRNAs function as suitable cancer biomarkers
- CCAT1e is an established eRNA in colorectal cancer. Did you find that in your dataset? Else include that and give reference.
Reviewer 2 Report
The authors used a machine learning approach to a CRC cohort in TCGA database to identify an eRNA signature for recurrence prediction. The results demonstrated good prediction accuracy, which may be applicable in future clinic use if more evidence is provided in future independent studies.
Major concerns:
1) A demographic table of the whole study cohort from the TCGA database with details will be very helpful and appreciated, including the training and testing sets, their distribution of age, gender, and TNM stage. While the selection for cohorts was random, it is good to show little bias existing in the study with the table, in addition to Fig. 1.
2) While the prediction signature is a set eRNAs, 22 for CC and 19 for RC, not a single eRNA, I think the section 2.5 and Fig. 6 have less significant contribution to the topic. However, it reminds me if there have been any studies on identification of gene signature (a set of coding genes) for CRC recurrence prediction. If yes, it will be of greater interest to compare these two signatures.
3) Different from the authors’ viewpoint, from Fig.7A, my reading is that myeloid leukocyte mediated immunity, centriole assembly, and DNA methylation are more related to CC eRNA signature; and regulation of DNA replication, ncRNA metabolic process and mRNA processing are more related to RC eRNA signature.
4) Since ENSR00000021608 is shared between CC and RC signatures, it should have been of more interest to dig on its regulated gene(s) and if this single eRNA being used as a signature for CRC, instead of eRNAs picked in section 2.5 and Fig. 6.
Minor concerns:
1) When “Several studies have reported”, no citation was given;
2) When abbreviation shown at the 1st time, no full name was given;
3) Some errors in sentences and grammar.
